# Impact of Insulin Therapies on Cancer Incidence in Type 1 and Type 2 Diabetes: A Population-Based Cohort Study in Reggio Emilia, Italy

**DOI:** 10.3390/cancers14112719

**Published:** 2022-05-31

**Authors:** Massimo Vicentini, Paola Ballotari, Francesco Venturelli, Marta Ottone, Valeria Manicardi, Marco Gallo, Marina Greci, Mirco Pinotti, Annamaria Pezzarossi, Paolo Giorgi Rossi

**Affiliations:** 1Epidemiology Unit, Azienda USL-IRCCS di Reggio Emilia, 42122 Reggio Emilia, Italy; paola.ballotari@ats-valpadana.it (P.B.); francesco.venturelli@ausl.re.it (F.V.); marta.ottone@ausl.re.it (M.O.); annamaria.pezzarossi@ausl.re.it (A.P.); paolo.giorgirossi@ausl.re.it (P.G.R.); 2Medical Diabetologist Association Coordinator, Diabetologist, 42122 Reggio Emilia, Italy; manicardivaleria@gmail.com; 3Endocrinology and Metabolic Diseases Unit, AO SS Antonio e Biagio e Cesare Arrigo of Alessandria, 15121 Alessandria, Italy; marco.gallo@ospedale.al.it; 4Primary Health Care Department, Azienda USL-IRCCS di Reggio Emilia, 42122 Reggio Emilia, Italy; marina.greci@ausl.re.it; 5Risk Management Team, Azienda USL-IRCCS di Reggio Emilia, 42122 Reggio Emilia, Italy; mirco.pinotti@ausl.re.it

**Keywords:** insulin, cancer, diabetes, glucose-lowering therapies

## Abstract

**Simple Summary:**

The aim of this population-based study was to assess the impact of insulin treatment on cancer incidence in subjects with type 1 or type 2 diabetes in Italy. We found that insulin use was associated with a 20% excess for all sites cancer incidence among people with type 2 diabetes, while people with type 1 diabetes did not show any excess. Liver, pancreatic, bladder, and neuroendocrine cancers seem to be the sites with strongest association.

**Abstract:**

Objective: To assess the effect of insulin on cancer incidence in type 1 (T1DM) and type 2 diabetes (T2DM). Methods: The cohort included all 401,172 resident population aged 20–84 in December 2009 and still alive on December 2011, classified for DM status. Drug exposure was assessed for 2009–2011 and follow up was conducted from 2012 to 2016 through the cancer registry. Incidence rate ratios (IRRs) were computed for all sites and for the most frequent cancer sites. Results: among residents, 21,190 people had diabetes, 2282 of whom were taking insulin; 1689 cancers occurred, 180 among insulin users. The risk for all site was slightly higher in people with T2DM compared to people without DM (IRR 1.21, 95% CI 1.14–1.27), with no excess for T1DM (IRR 0.73, 95% CI 0.45–1.19). The excess in T2DM remained when comparing with diet-only treatment. In T2DM, excess incidence was observed for liver and pancreas and for NETs: 1.76 (95% CI 1.44–2.17) and 1.37 (95% CI 0.99–1.73), respectively. For bladder, there was an excess both in T1DM (IRR 3.00, 95% CI 1.12, 8.02) and in T2DM (IRR1.27, 95% CI 1.07–1.50). Conclusions: Insulin was associated with a 20% increase in cancer incidence. The risk was higher for liver, pancreatic, bladder and neuroendocrine tumours.

## 1. Introduction

### 1.1. Rationale

Patients with type 2 diabetes (T2DM), have an increased risk of cancer [1,2]. Studies with different design and in different populations showed an increased risk for breast [2,3,4], endometrial [2,5,6], pancreatic, colorectal [2,4,7], and liver cancer [2,7]. However, whether the relationship between diabetes and cancer is direct or mediated through biological mechanism, like insulin resistance and hyperinsulinemia, or it is correlated to common risk factors, such as obesity and metabolic syndrome, remains unclear [8]. Insulin is a growth factor and it is possible that high levels of endogenous or administration of exogenous insulin could stimulate tumour growth [9,10]. The potential oncogenic role of some insulin analogues (glargine in particular) has been suggested because of the known role in cellular reproduction of insulin-like growth factor (IGF-1) and insulin receptor signalling pathways [11]. Insulin is both a metabolic hormone and a growth factor, similar to its cognate factor IGF-1. Cancer cells may be more responsive to the mitogenic effect of insulin overexpressing the insulin receptor (IR) and, specifically, IR isoform A. Furthermore, at supraphysiological concentrations (e.g., obesity and hyperinsulinemia compensatory to insulin resistance), insulin can also interact with the IGF1 receptor. High insulin levels also reduce IGF-1-binding proteins and increase free-IGF-1, overactivating the mitogenic effects of the IGF-1 pathway in tumour tissues [12]. Moreover, in cancer cells, glargine and long-acting insulins have been shown to exert a greater proliferative effect relative to human insulin both through the IGF1 receptor and IR isoform A [13].

The main targets of insulin are liver, adipose tissue, skeletal muscle, and brain, but the insulin receptor is ubiquitous and is also expressed in the heart, lung, pancreas, kidney, placenta, vascular endothelium, monocytes, granulocytes, fibroblast, and erythrocytes [14]. Since 2009, data from observational studies suggest an association between insulin use and cancer risk, but results are often conflicting and inconclusive [15,16,17]. Evidence from randomized controlled trial are also limited. Two meta-analyses published in 2009 did not find an increased risk of cancer using insulin glargine [18,19], but these studies were small and with a short follow-up. A systematic review published in 2013 showed a significant increase or decrease in cancer risk for insulin users, depending on cancer site. Insulin exposure was associated with an increase in pancreatic, liver, kidney, colorectal, stomach, and lung cancers, but a decrease in risk for prostate cancer. However, few studies from pooled analyses were available, and a subanalysis of possible determinants of cancer risk was therefore not feasible. Moreover, the results of each study showed substantial variation in reported cancer risk [8]. To assess the duration of insulin use, treatment switch and exposure duration are fundamental. In particular, an increase in pancreatic cancer risk must be carefully evaluated to avoid reverse causality. The CARING study, an observational study following 327,112 insulin users from five European countries, showed no consistent evidence in the cancer risk associated with insulin glargine or detemir use compared to human insulin use. Although the authors observed increased and decreased cancer risk for some sites in glargine users, no trends in the risk with duration of treatment were seen.

Switching from oral glucose-lowering drugs to insulin or from human insulin to glargine insulin may be a sign of failure in glucose target level control. Poor glycaemic control may be, in turn, a sign of underlying cancer and thus an association between cancer incidence and use of insulin in the short term could be due to reverse causality [20]. The role of undiagnosed cancer on the likelihood of diabetes diagnosis has been suggested by evidence from a Danish population-based study; the authors reported a higher cancer incidence rate ratio in the first year after diabetes diagnosis, which then decreased with diabetes duration [21]. A similar trend was observed considering the start and duration of insulin therapy, with an incidence rate ratio starting at 5 and decreasing to 1.3 after 5 years of insulin treatment [21].

For type 1 diabetes (T1DM), evidence of an effect of insulin on cancer incidence is limited and variable. Cohort studies have shown a 10–37% increased risk for all cancers, but case-controls studies have shown no association. Because these studies usually have a small sample size, they are not powered enough to explore site-specific cancer incidence. However, data suggest an increased risk for pancreatic, liver, and stomach cancer [22]. These results are consistent with evidence from a large population-based study that included data from 5 national diabetes and cancer registries [23]. In addition, Carstensen et al. showed an increased risk for endometrial and kidney cancer and highlighted differences in cancer risk by sex (i.e., higher in women) and by diabetes duration, (i.e., higher in the first year after diagnosis, then decreasing over time) [23]. Differences in study design and quality of data sources may explain the heterogeneity of the results. Dose and duration of insulin therapy, diabetes history (duration, date of diagnosis, and progression) may be misclassified; furthermore, changes in glucose-lowering medications and combined glucose-lowering medications are difficult to account, thereby possibly introducing appreciable biases [24].

Several case-control studies and some meta-analyses indicate diabetes as a potential risk factor for the development of neuroendocrine tumours (NETs), especially for non-functioning tumours of pancreatic or gastric origin [25]. However, the mechanisms linking diabetes to NETs development, if any, are largely unknown. As well as all organs and tissues, diabetes may worsen chronic inflammation and intracellular oxidative stress, leading to DNA mutation. However, diabetes has also been proposed as an early paraneoplastic condition or a consequence of a NET-induced impairment of glucose metabolism, rather than a real factor promoting tumour initiation. Finally, some of the newest drugs used to treat diabetes (e.g., glucagon-like peptide-1 (GLP-1) receptor agonists and dipeptidyl peptidase-4 (DPP-4) inhibitors) may exert some promoting effects on the development of NETs. These suggested effects would be linked to their potential regenerative influence on pancreatic cells, both of exocrine and of endocrine origin [26].

### 1.2. Aim

The aim of this population-based cohort study was to assess the impact of insulin treatment on cancer incidence in subjects with type 1 (T1DM) or type 2 diabetes (T2DM) in Italy.

## 2. Research Design and Methods

### 2.1. Data Sources

Population-based Diabetes Registry and Population-based Cancer Registry. These data sources have been linked to build a population-based cohort allowing the study the associations between diabetes and cancers. The process to build the cohort has been previously described [27,28].

### 2.2. Study Population

The Reggio Emilia province, in Northern Italy, has about 530,000 resident inhabitants. The Cohort included the 401,172 inhabitants resident at 31 December 2009 and still alive and resident on 31 December 2011. Through record linkage with the diabetes registry (accessed at November 2017) the population was classified as with or without diabetes on 31 December 2009, ref. [28] distinguishing between Type 1 and Type 2 diabetes. Diseases of the exocrine pancreas, drug-induced diabetes (121), were excluded from the analysis, while women with only gestational diabetes were considered in the population without diabetes.

As previously reported [27], the Diabetes Registry includes information about people with diabetes collecting data from six routinely collected information systems: hospital discharge, drug dispensation, biochemistry laboratory (for glycated haemoglobin), disease-specific exemption, diabetes outpatient clinic activity, and mortality. Diagnoses are checked by a diabetologist or other physician to exclude people receiving glucose lowering therapies for reasons other than diabetes and women with gestational diabetes.

### 2.3. Patient Characteristics and Outcomes

The exposure to insulin was measured for the period 2009–2011, while cancer incidence was followed up from 1 January 2012 to 31 December 2016 (Figure 1). This strict distinction between exposure assessment period, lasting at least two years, and follow-up period minimizes the risk of reverse causality bias, whether due to ascertainment or to protopathic bias [20]. The main comparison was between T1DM and T2DM pancreatic diabetes, although this analysis could not be adjusted for different baseline risk.

To minimize this bias, we also compared only T2DM patients exposed to insulin (alone or in any combination) with those on diet-only regimen during the same period. This analysis permitted also adjusting for the duration of diabetes.

The procedure to assess the therapy of the patients with diabetes has been described before [28]. Briefly, data routinely collected by the Pharmacy Drug Dispensation (AFT) and Hospital Direct Drug Dispensation (FED) databases were used. Patients were classified in three groups according to drug prescriptions received in 2009–2012: (1) insulin alone or in any combination (ATC code = A10A* and A10B*); (2) insulin alone (ATC code = A10A* and not A10B*), and (3) untreated (diet only) (only for T2DM) [Appendix A]. Subjects with DM were classified as “drug consumers” if they had had at least 2 prescriptions per year of the drugs falling in the same group.

The resident population was followed up for 5 years from on 1 January 2012 to 31 December 2016 or cancer diagnosis, death, emigration (Figure 1). Vital status and migration information were collected from the Civil Registry Office. Follow-up began.

The outcome of interest was cancer incidence. Only the first cancer of each site in the case of multiple tumours during follow up was considered. Cancer site was coded according to International Statistical Classification of Diseases and Related Health Problems, 10th Revision (ICD-10) (see Appendix A for major tumor sites). For neuroendocrine tumours we modified the list of ICD-O-3 codes used in the Rarecare project [29] adding all lung carcinoid tumours (see Appendix A for the details). Non-melanoma skin cancers (C44), chronic myeloproliferative disorders, and myelodysplastic syndromes (D45–D47) are excluded from incidence.

### 2.4. Statistical Analyses

To compare cancer incidence between people with DM with different DM type and on different therapies regimens, we report incidence rate ratios (IRRs) with relative 95% confidence intervals (95% CI) computed with multivariate Poisson regression model for all-sites and for specific investigated sites (lung, kidney, lymphomas, bladder, stomach, corpus uteri, ovary, breast, liver, colorectal, prostate, pancreas, and other sites). Specific model for neuroendocrine tumours was also carried out. We report two types of models: in the first we used subjects without diabetes as reference group, in the second we used untreated patients with T2DM (i.e., in diet-only programmes) as reference group. In this study, we do not present formal set of hypothesis, we did not define a threshold of significance; therefore, confidence intervals and *p* values should be interpreted as the measure of the likelihood that the observed differences were due to chance.

All models were adjusted for age at baseline, citizenship, and sex. For models including only peoples with DM, diabetes duration was available and thus models were adjusted also for this variable [Appendix A].

## 3. Results

### 3.1. Participants

The study cohort included 758 people with a diagnosis of type 1 diabetes, 21,190 with type 2 diabetes, and 379,103 people with no diagnosis of diabetes mellitus as of 31 December 2009, and all still alive on 31 December 2011; 121 people with secondary diabetes were excluded. The three groups differed in terms of age: people with T2DM were older than people with T1DM and the general population.

There were 2282 T2DM insulin users, of whom 1332 used insulin alone.

### 3.2. Outcome—All Sites

In people with T1DM, 16 cancer cases were diagnosed; in people with T2DM, 1689 new cancer cases were diagnosed during follow-up (180 in insulin users, 121 of whom were insulin-only users), while 12,882 cancers were diagnosed in people without diabetes (Table 1 and Table 2).

The risk of all sites incidence in T1DM patients was lower than in the general population (IRR = 0.73, 95% CI 0.45, 1.19). The risk for T2DM insulin users was increased by 21% compared to the general population, (IRR = 1.21, 95% CI 1.04, 1.40); the excess risk was slightly higher in the subgroup of insulin-only users (IRR = 1.35, 95% CI 1.12, 1.61). The excess risk remained when comparing insulin users with the group of T2DM patients on diet-only therapy (Table 2).

#### 3.2.1. Stomach

No case of stomach cancer was detected in T1DM patients. Non excess risk was observed risk for this cancer in T2DM insulin users, compared to the general population (IRR = 0.8, 95% CI 0.35, 1.8).

#### 3.2.2. Colorectal Cancer

Only one colorectal cancer was observed in T1DM patients, compared with the two expected. The risk for T2DM insulin users was similar to that of the general population (IRR 1.13, 95% CI 0.7, 1.7). Increased risk, compatible with random fluctuation, was observed for those taking insulin alone compared to patients with T2DM treated with diet alone (IRR = 1.67, 95% CI: 0.76, 3.67) (Table 3, Table 4, Table 5 and Table 6).

#### 3.2.3. Liver

Only one liver cancer was observed in T1DM patients, while 0.5 were expected. The risk of liver cancer in T2DM insulin users was higher than in the general population (IRR = 5.06, 95% CI 3.24, 7.90); the excess risk was similar when comparing those using insulin alone with those treated with diet alone (IRR = 4.52, 95% CI 1.89, 10.82) (Table 3).

#### 3.2.4. Pancreas

No case of pancreatic cancer was detected in T1DM patients. The risk was higher among T2DM insulin users than in the general population (IRR = 2.4, 95% CI 1.5, 3.9); the excess risk was similar when comparing those taking insulin alone to patients with T2DM treated with diet alone (IRR = 2.32, 95% CI 0.85, 6.29) (Table 3).

#### 3.2.5. Bladder

Compared to the general population, an increased risk was observed for patients with T1DM (IRR = 3.00, 95% CI 1.12, 8.02) and a slight increased risk was observed for T2DM in insulin users (IRR = 1.5, 95% CI 1.0, 2.3) and when comparing insulin-only users to patients with T2DM treated with diet alone (IRR = 1.58, 95% CI 0.72, 3.48) (Table 3). Both excesses may have been due to random fluctuations.

#### 3.2.6. Neuroendocrine Tumours (NETs)

No case of NETs was registered among T1DM patients. The risk of developing NET (all) for T2DM insulin users, compared to the general population, was increased (IRR = 1.4, 95% CI 0.7, 3.1). For insulin in any combination, the risk was IRR = 3.44 (95% CI 1.04, 11.38) compared to patients with T2DM treated with diet alone, while no case was found in insulin-alone therapy (Table 7).

#### 3.2.7. Other Sites

No excesses of cancer risk emerged for trachea, bronchus, and lung, breast, corpus uteri, ovary, prostate, kidney and lymphomas (see Table 3).

Furthermore, compared to general population, no increased risk for T1DM patients (IRR = 0.31, 95% CI 0.08, 1.24) or for T2DM insulin users (IRR = 0.90, 95% CI 0.64, 1.28) was observed in the group of miscellanea cancer sites (Appendix A).

## 4. Discussion

### 4.1. Key Results

Patients with type 2 diabetes treated with insulin showed a 20% excess risk for all cancer sites compared with both the general population and with patients with diabetes on diet-only treatment. On the contrary, taking in mind the limits due to small numbers, no excess risk was appreciable for patients with type 1 diabetes.

Despite not having enough power to observe any difference between the effect of insulin alone or in combination with other drugs. Compared to type 2 patients on diet only, the excess risk for all sites was almost exclusively due to an excess in a few sites: pancreas, liver, bladder, corpus uteri, ovary, and colon-rectum. Nevertheless, the excess in the last three sites was compatible with random fluctuations. The other sites for which we observed an excess incidence for the population with diabetes did not show any excess in those on insulin. Furthermore, aggregating neuroendocrine tumours in different sites, we observed a strong excess for people with type 2 diabetes treated with insulin, albeit with the limitation of the relatively small number of patients.

Although we did not find any excess in type 1 diabetes for all sites, we observed an increased risk for bladder cancer.

### 4.2. Interpretation

These results are in line with those of previously published studies, which found a slight excess in cancer incidence in people with diabetes compared with the general population [7,30,31,32]. The effect seems to increase with the duration of the diabetes and the severity of hyperglycaemic status over the course of life [33,34]. Therefore, any excess observed in insulin users may be confounded because insulin use in type 2 diabetes patients is associated with duration and severity of the disease, thus explaining the excess compared with the population with type 2 diabetes on diet only.

Any hypothesis of a direct effect of insulin on cancer incidence in type 2 diabetes should explain why there is no excess in patients with type 1 diabetes, who surely have an average longer use of insulin than do T2DM patients. In the specific case of the three sites that could have a direct effect, it is worth noting that an excess in bladder cancer was detectable also in the type 1 diabetes group, while for pancreas and liver, the study had insufficient power to observe even a large effect. Pooled analyses of five large cohorts of T1DM patients showed an increase in risk of about 50% for liver and pancreas but no increase for bladder cancer [23]. It is worth noting that the overall occurrence of bladder cancer was much lower in the pooled cohorts than in our population and that a large variability between registries and periods has been observed for this cancer due to the issue of including all in situ neoplasms among incident cases, as international registration rules require.

With regards to this issue, even if most T1DM and T2DM patients are exposed for decades to increasing insulin concentrations, it should be emphasized that the two conditions are very different. Patients with insulin resistance, obesity, and/or T2DM are exposed over the long term to high levels of pancreas-secreted insulin, which first passes through the liver (first passage insulin). In the liver, a high percentage of exogenous insulin is retained and degraded, whereas the remaining aliquot reaches the peripheral tissues through the systemic circulation [35]. Conversely, people with T1DM (as well as subjects who undergo total pancreatectomy or with severe pancreatic disease) totally depend on exogenous insulin administration, which arrives to the liver and to peripheral tissues simultaneously and at a similar concentration. Therefore, the liver/peripheral tissue insulin concentration ratio is much higher in people with T2DM than in those with T1DM. The relative liver hyperinsulinemia observed in T2DM, together with excess unused substrates (i.e., glucose) and other hormone abnormalities, has been suggested to contribute to the excess risk in liver cancer.

As to the prostate, diabetes seems to exert a protective effect against cancer by lowering circulating levels of testosterone [36]. This effect on endogenous androgen levels could prevail on the mitogenic effects of hyperinsulinemia and insulin therapy.

### 4.3. Strengths and Limitations

This is a population-based study including all the resident population and using information from a well-established diabetes registry that uses six different data sources. The registry also makes it possible to identify patients with type 2 diabetes treated with diet only or untreated; all diagnoses are confirmed through clinical records check. The accuracy of the registration has been validated in previous studies [7].

In our study design, as we kept the exposure definition phase and follow-up phase strictly separate, we should have almost entirely eliminated the possibility of any reverse causality bias and immortal time bias [37]. Furthermore, given that we compared insulin users with those on diet-only treatment, the risk of misclassification was quite small since few patients shift directly from diet to insulin. On the other hand, the exposure to insulin was assessed through administrative databases using algorithms that may misclassify some users and definitely cannot measure the duration of exposure. Fortunately, there is almost no purchase of insulin outside of the Italian NHS system channels. Also, the date of diagnosis, and therefore the duration of disease, may be inaccurate, particularly for those cases that were prevalent when we started registration.

The absence of a clear tumorigenic effect of insulin therapy in our study could be partially due to the relatively high prevalence of people using metformin together with insulin, as the biguanide has been steadily supposed to play a protective effect against breast cancer and other neoplasms, especially by observational studies. However, we could not find any evidence of an antitumor effect of metformin in a previously published analysis on this same cohort of patients [28].

Due to the relatively small number of events and to the short follow-up, our study had no power to detect differences in site-specific cancer incidence and in T1DM. Pooling data with other population-based registries and/or extending the follow-up could overcome this issue.

Another important limitation of our study is the lack of information about the type of insulin used (it may be particularly interesting to distinguish between the effect of human vs. analogue insulin and glargine vs. detemir), the mean daily insulin doses, and patients’ glyco-metabolic control [38].

Furthermore, our registries cannot provide data on many other important risk factors (e.g., sociodemographic characteristics, lifestyle, smoking, familiarity for cancer, work exposure to toxic agents, and/or compliance with cancer screening tests) as well as for other potential confounders (e.g., the use of other drugs, such as aspirin or statins). Uncontrolled behavioural risk factors, i.e., smoking and alcohol consumption, should not be the cause of the observed excess since the cancer sites that are more directly related with these risk factors, such as lung and oesophagus, do not show any excess. We also do not have any information on, and thus cannot adjust for, body mass index (BMI). Nevertheless, the association between BMI or other metabolic risk factors and liver [39], pancreatic, or bladder cancer is much weaker than the observed excess in our cohort. Finally, our data refer to a population with a majority of white Caucasian subjects and cannot thus be extrapolated to other ethnic groups.

## 5. Conclusions

In our cohort, insulin use was associated with a 20% excess for all sites cancer incidence among people with type 2 diabetes, while this excess was not appreciable in people with type 1 diabetes. Liver, pancreatic, bladder, and neuroendocrine cancers seem to be the sites with strongest association. For bladder cancer, an excess was present also for people with type 1 diabetes.

## Figures and Tables

**Figure 1 cancers-14-02719-f001:**
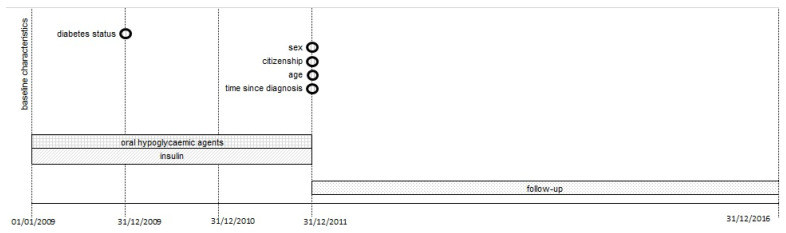
Study timeline showing the exposure assessment follow up periods.

**Table 1 cancers-14-02719-t001:** Baseline characteristics of the cohort according to the presence of type 1 and type 2 diabetes (DM1; DM2) as of 31 December 2009 and still alive at 31 December 2011, outcomes and follow-up completeness (period 2011–2016). Reggio Emilia 2011-16, resident population aged 20–84 years.

Characteristics	Population with DM1 (N = 758)		Population with DM2 (N = 21,190)		Population without DM (N = 379,103)	
at Baseline	Mean	SD	Mean	SD	Mean	SD
**age**	46.1	14.6	65.8	11.4	47.6	16.4
age class	N	%	N	%	N	%
<40	279	36.8	478	2.3	139,320	36.8
40–49	195	25.7	1535	7.2	82,802	21.8
50–59	139	18.3	3715	17.5	59,657	15.7
60–69	84	11.1	6391	30.2	48,133	12.7
70–79	55	7	6823	32.2	36,579	9.7
80+	8	1.1	2248	10.6	12,612	3.3
sex						
male	410	52.9	11,823	55.8	185,144	48.8
female	357	47.1	9367	44.2	193,959	51.2
**citizenship**						
Italian	671	88.5	19,570	92.3	325,440	85.8
Foreign national	87	11.4	1620	7.6	53,663	14.2
during follow up	N	%	N	%	N	%
person-years	3665		94,429		1,836,603	
deaths 2012–2016	35	4.6	3599	17.0	14,630	3.9
relocations 2012–2016	0	-	14	0.1	714	0.2
cancers	16	2.1	1689	8.0	12,882	3.4

**Table 2 cancers-14-02719-t002:** Number, incidence rates ratios (IRR), and 95% confidence intervals (95% CI) of cancer by diabetes status and type and by therapy. Reggio Emilia 2011-16, resident population aged 20–84 years.

Categories	All Sites							
	Patients with DM2 in Diet-Only as Reference				Population without DM as Reference			
	N	IRR	95% CI		N	IRR	95% CI	
Population with no DM (N = 379,103)	-	-	-	-	12,882	1.00	-	-
Population with DM (N = 21,948)	-	-	-	-	1705	1.20	1.14	1.26
1. Type 1 diabetes (N = 758)	-	-	-	-	16	0.73	0.45	1.19
2. Type 2 diabetes (N = 21,190)	-	-	-	-	1689	1.21	1.14	1.27
2.1 Insulin alone or in any combination (N = 2282)	180	1.19	0.94	1.51	180	1.21	1.04	1.40
2.2 Insulin alone (N = 1332)	121	1.38	1.06	1.81	121	1.35	1.12	1.61
2.3 Untreated (diet only) (N = 4077)	301	1.00	-	-	301	1.09	0.97	1.22

**Table 3 cancers-14-02719-t003:** Number, incidence rates ratios (IRR), and 95% confidence intervals (95% CI) of colorectal, pancreatic, liver, trachea, bronchus and lung cancer, stomach cancer and lymphoma, corpus uteri, breast (women only), ovary, kidney, bladder, and prostate cancer by type of treatment with type 1 and type 2 diabetes (DM1; DM2). Reggio Emilia 2011-16, resident population aged 20–84 years. See Appendix A for “other sites”.

	Colon-Rectum (C18–C20)								Pancreas (C25)								Liver (C22)							
	Patients with DM2 in Diet-Only as Reference				Population without DM as Reference				Patients with DM2 in Diet-Only as Reference				Population without DM as Reference				Patients with DM2 in Diet-Only as Reference				Population without DM as Reference			
	N	IRR	95% CI		N	IRR	95% CI		N	IRR	95% CI		N	IRR	95% CI		N	IRR	95% CI		N	IRR	95% CI	
Population with no DM (N = 379,103)					1279	1.00	-	-					501	1.00	-	-					309	1.00	-	-
Population with DM (N = 21,948)					197	1.18	1.01	1.38					115	1.74	1.42	2.14					102	2.57	2.04	3.23
1. Type 1 diabetes (N = 758)					1	0.51	0.07	3.61					0	-	-	-					1	2.00	0.28	14.30
2. Type 2 diabetes (N = 21,190)					196	1.19	1.02	1.39					115	1.76	1.44	2.17					101	2.57	2.04	3.24
2.1 Insulin alone or in any combination (N = 2282)	20	1.32	0.64	2.74	20	1.13	0.72	1.76	17	1.72	0.70	4.21	17	2.39	1.47	3.88	21	3.28	1.46	7.37	21	5.06	3.24	7.90
2.2 Insulin alone (N = 1332)	16	1.67	0.76	3.67	16	1.45	0.88	2.38	12	2.32	0.85	6.29	12	2.70	1.52	4.79	16	4.52	1.89	10.82	16	6.18	3.72	10.27
2.3 Untreated (diet only) (N = 4077)	28	1.00	-	-	28	0.86	0.59	1.25	13	1.00	-	-	13	1.00	0.58	1.74	13	1.00	-	-	15	1.90	1.13	3.20

**Table 4 cancers-14-02719-t004:** Number, incidence rates ratios (IRR), and 95% confidence intervals (95% CI) of colorectal, pancreatic, liver, trachea, bronchus, and lung cancer, stomach cancer and lymphoma, corpus uteri, breast (women only), ovary, kidney, bladder, and prostate cancer by type of treatment with type 1 and type 2 diabetes (DM1; DM2). Reggio Emilia 2011-16, resident population aged 20–84 years. See Appendix A for “other sites”.

	Trachea, Bronchus, and Lung Cancer (C33–C34)								Lymphoma (C81–C85, C96)								Stomach (C16)							
	Patients with DM2 in Diet-Only as Reference				Population without DM as Reference				Patients with DM2 in Diet-Only as Reference				Population without DM as Reference				Patients with DM2 in Diet-Only as Reference				Population without DM as Reference			
	N	IRR	95% CI		N	IRR	95% CI		N	IRR	95% CI		N	IRR	95% CI		N	IRR	95% CI		N	IRR	95% CI	
Population with no DM (N = 379,103)					1414	1.00	-	-					582	1.00	-	-					503	1.00	-	-
Population with DM (N = 21,948)					247	1.24	1.08	1.42					68	1.14	0.86	1.43					91	1.29	1.02	1.61
1. Type 1 diabetes (N = 758)					0	-	-	-					2	1.94	0.48	7.80					0	-	-	-
2. Type 2 diabetes (N = 21,190)					247	1.25	1.09	1.44					66	1.09	0.84	1.41					91	1.30	1.04	1.63
2.1 Insulin alone or in any combination (N = 2282)	23	1.37	0.71	2.64	23	1.11	0.73	1.67	3	0.67	0.15	3.01	3	0.48	0.15	1.48	6	1.15	0.34	3.89	6	0.80	0.35	1.79
2.2 Insulin alone (N = 1332)	11	0.78	0.40	1.84	11	0.83	0.46	1.51	3	1.13	0.25	5.10	3	0.79	0.25	2.46	4	1.20	0.29	4.90	4	0.84	0.31	2.26
2.3 Untreated (diet only) (N = 4077)	39	1.00	-	-	39	0.99	0.72	1.37	16	1.00	-	-	16	1.34	0.81	2.21	14	1.00	-	-	14	1.00	0.59	1.71

**Table 5 cancers-14-02719-t005:** Number, incidence rates ratios (IRR), and 95% confidence intervals (95% CI) of colorectal, pancreatic, liver, trachea, bronchus, and lung cancer, stomach cancer and lymphoma, corpus uteri, breast (women only), ovary, kidney, bladder, and prostate cancer by type of treatment with type 1 and type 2 diabetes (DM1; DM20). Reggio Emilia 2011-16, resident population aged 20–84 years. See Appendix A for “other sites”.

	Corpus Uteri (C54)								Breast (C50)								Ovary (C56)							
	Patients with DM2 in Diet-Only as Reference				Population without DM as Reference				Patients with DM2 in Diet-Only as Reference				Population without DM as Reference				Patients with DM2 in Diet-Only as Reference				Population without DM as Reference			
	N	IRR	95% CI		N	IRR	95%CI		N	IRR	95% CI		N	IRR	95% CI		N	IRR	95% CI		N	IRR	95% CI	
Population with no DM (N = 379,103)					333	1.00	-	-					2056	1.00	-	-					200	1.00	-	-
Population with DM (N = 21,948)					41	1.49	1.06	2.08					145	0.97	0.81	1.15					25	1.56	1.02	2.39
1. Type 1 diabetes (N = 758)					0	-	-	-					4	1.13	0.42	3.01					0	-	-	-
2. Type 2 diabetes (N = 21,190)					41	1.52	1.09	2.13					141	0.96	0.81	1.15					25	1.60	1.04	2.46
2.1 Insulin alone or in any combination (N = 2282)	5	2.64	0.57	12.35	5	1.60	0.66	3.88	14	0.85	0.38	1.90	14	0.84	0.50	1.43	3	0.97	0.16	5.72	3	1.69	0.54	5.33
2.2 Insulin alone (N = 1332)	3	3.29	0.55	19.80	3	1.68	0.58	5.27	8	0.83	0.31	2.36	8	0.86	0.43	1.73	2	1.42	0.18	11.16	2	2.00	0.50	8.12
2.3 Untreated (diet only) (N = 4077)	5	1.00	-	-	5	0.94	0.39	2.28	30	1.00	-	-	30	1.05	0.73	1.50	6	1.00	-	-	6	1.97	0.86	4.46

**Table 6 cancers-14-02719-t006:** Number, incidence rates ratios (IRR), and 95% confidence intervals (95% CI) of colorectal, pancreatic, liver, trachea, bronchus, and lung cancer, stomach cancer and lymphoma, corpus uteri, breast (women only), ovary, kidney, bladder, and prostate cancer by type of treatment with type 1 and type 2 diabetes (DM1; DM2). Reggio Emilia 2011-16, resident population aged 20–84 years. See Appendix A for “other sites”.

	Kidney (C64)								Bladder (C67, D09, D41.4)								Prostate (C61)							
	Patients with DM2 in Diet-Only as Reference				Population without DM as Reference				Patients with DM2 in Diet-Only as Reference				Population without DM as Reference				Patients with DM2 in Diet-Only as Reference				Population without DM as Reference			
	N	IRR	95% CI		N	IRR	95% CI		N	IRR	95% CI		N	IRR	95% CI		N	IRR	95% CI		N	IRR	95% CI	
Population with no DM (N = 379,103)					434	1.00	-	-					894	1.00	-	-					1172	1.00	-	-
Population with DM (N = 21,948)					57	1.10	0.83	1.46					168	1.28	1.09	1.52					160	0.91	0.77	1.08
1. Type 1 diabetes (N = 758)					0	-	-	-					4	3.00	1.12	8.02					3	1.59	0.51	4.95
2. Type 2 diabetes (N = 21,190)					57	1.11	0.84	1.48					164	1.27	1.07	1.50					157	0.91	0.78	1.08
2.1 Insulin alone or in any combination (N = 2282)	5	0.92	0.22	3.75	5	0.94	0.39	2.27	20	1.13	0.56	2.29	20	1.50	0.96	2.33	16	0.89	0.42	1.89	16	0.92	0.57	1.52
2.2 Insulin alone (N = 1332)	1	0.42	0.04	4.73	1	0.30	0.04	2.16	14	1.58	0.72	3.48	14	1.64	0.97	2.78	13	1.20	0.53	2.70	13	1.17	0.68	2.02
2.3 Untreated (diet only) (N = 4077)	8	1.00	-	-	8	0.79	0.39	1.59	35	1.00	-	-	35	1.35	0.97	1.90	35	1.00	-	-	35	1.02	0.73	1.43

**Table 7 cancers-14-02719-t007:** Number, incidence rates ratios (IRR), and 95% confidence intervals (95% CI) of neuroendocrine cancer, all sites and only from pancreas and digestive tract, by type of treatment with type 1 and type 2 diabetes (DM1; DM2), aged 20–84 years. See Appendix A for “neuroendocrine tumours in other sites” and Appendix A site and morphology code selection [29].

	Neuroendocrine Carcinoma (all)								Neuroendocrine Carcinoma (only Pancreas and Digestive Tract)							
	Patients with DM2 in Diet-Only as Reference				Population without DM as Reference				Patients with DM2 in Diet-Only as Reference				Population without DM as Reference			
	N	IRR	95% CI		N	IRR	95% CI		N	IRR	95% CI		N	IRR	95% CI	
Population with no DM (N = 379,103)					381	1.00	-	-					113	1.00	-	-
Population with DM (N = 21,948)					59	1.29	0.97	1.70					11	0.95	0.31	1.79
1. Type 1 diabetes (N = 758)					0	-	-	-					0	-	-	-
2. Type 2 diabetes (N = 21,190)					59	1.37	0.99	1.73					11	0.97	0.51	1.82
2.1 Insulin alone or in any combination (N = 2282)	7	3.44	1.04	11.38	7	1.44	0.68	3.05	0	-	-	-	0	-	-	-
2.2 Insulin alone (N = 1332)	0	-	-	-	0	-	-	-	0	-	-	-	0	-	-	-
2.3 Untreated (diet only) (N = 4077)	9	1.00	-	-	9	0.99	0.51	1.93	0	1.00	-	-	0	-	-	-

## Data Availability

Individual anonymized data are available presenting a motivated request from the authors, subject to Ethic Committee authorization. Aggregated data are available from the authors on request. Requests should be addressed to info.epi@ausl.re.it.

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
