# Peer review of "Impact of Insulin Therapies on Cancer Incidence in Type 1 and Type 2 Diabetes: A Population-Based Cohort Study in Reggio Emilia, Italy"

_cancers, 2022, doi:10.3390/cancers14112719_

Round 1
Reviewer 1 Report
The authors have shown in the present study that insulin therapy was associated with a 20% excess for all sites of cancer incidence among people with type 2 diabetes in Italy, although other causes could not be ruled out. I have the following comments for improving the manuscript:
- In the abstract, the following fragment is unclear "all resident population 20–84" if you meant to age range, please rephrase.
- The following statement is unclear also " The procedure to assess the therapy of the patients with diabetes has been described 143 before [met]." What do you mean by "met"?
- Also, the following sentence is unclear "Briefly, dara routinely collected by..." Please correct "dara".
- In line 168, please correct the verb "in the first we uses subjects without diabetes."
- The references' style should be consistent for all, some of the references include doi, but most do not.
Author Response
Reviewer 1
The authors have shown in the present study that insulin therapy was associated with a 20% excess for all sites of cancer incidence among people with type 2 diabetes in Italy, although other causes could not be ruled out. I have the following comments for improving the manuscript:
We thank the reviewer for the suggestions.
- In the abstract, the following fragment is unclear "all resident population 20–84" if you meant to age range, please rephrase.
RE: thanks we rephrased the sentence.
- The following statement is unclear also " The procedure to assess the therapy of the patients with diabetes has been described 143 before [met]." What do you mean by "met"?
RE: it is a typo, it is a bibliographic citation. We corrected with the appropriate reference number.
- Also, the following sentence is unclear "Briefly, dara routinely collected by..." Please correct "dara".
RE: we corrected in “data”.
- In line 168, please correct the verb "in the first we uses subjects without diabetes."
RE: ok, we corrected in “used”.
- The references' style should be consistent for all, some of the references include doi, but most do not.
RE: we deleted doi in all references.
Reviewer 2 Report
This is a study of CA incidence in a population. It is well done but there are several issues:
- Re-do the ABSTRACT. you have to mention that the DM cohort was derived from a population of ~380,000. You should also compare cancer incidence to people without diabetes.
- INTRO - please rewrite lines 46-47 - it is not clear as written. Are you saying insulin works thru the IGF receptor?
- Line 144 - data
- Line 168 - used
- Line 169 - remove "and"
- Line 196 - remove "all insulin use group" - it is self-understood
- Table 2 - what is the 21,190 group? who are they?
- I do not think you have to present data in the text on each type of tumor. The reader can look at the tables by him/her-self. Rather just mention in type 1 DM xx tumor was more common; in T2DM yy tumors were more common, especially among insulin users, etc. As it stands the reading of the data is tedious.
- Your conclusions re T1DM are open to question. Age is one of the strongest risk factors for CA. The T1DM cohort is young, at an age that CA is not common. Please temper your remarks.... Please shorten the DISC section. It can be made tighter. ... Finally there were a lot of NET tumors. They are not that common in the population. Why so many here?
Author Response
Reviewer 2
- Re-do the ABSTRACT. you have to mention that the DM cohort was derived from a population of ~380,000. You should also compare cancer incidence to people without diabetes.
RE: we agree with the reviewer. We rephrased better explained how the cohort was built.
- INTRO - please rewrite lines 46-47 - it is not clear as written. Are you saying insulin works thru the IGF receptor?
RE: we agree with the reviewer. We completely rewrote the sentence and we added more appropriate referennces.
- Line 144 – data
RE: we corrected.
- Line 168 – used
RE: ok thank you.
- Line 169 - remove "and"
RE: ok thank you, corrected.
- Line 196 - remove "all insulin use group" - it is self-understood
RE: Ok, thank you.
- Table 2 - what is the 21,190 group? who are they?
RE: this is the group of patients with type 2 diabetes. We re-framed the label.
- I do not think you have to present data in the text on each type of tumor. The reader can look at the tables by him/her-self. Rather just mention in type 1 DM xx tumor was more common; in T2DM yy tumors were more common, especially among insulin users, etc. As it stands the reading of the data is tedious.
RE: We agree with the reviewer. We reduced the text.
- Your conclusions re T1DM are open to question. Age is one of the strongest risk factors for CA. The T1DM cohort is young, at an age that CA is not common. Please temper your remarks.... Please shorten the DISC section. It can be made tighter. ... Finally there were a lot of NET tumors. They are not that common in the population. Why so many here?
RE: We agree with the reviewer that our cohort has very low power for T1DM and age structure contribute to this lack of power. The structure of the population with T1DM is younger than that with T2DM, but it is not younger than that of the general population (see table 1). We added a sentence in the limit section and we tempered our remarks in the conclusions.
For neuroendocrine neoplasms, we reported the list of codes in the supplementary material tables 5-6 and we added appendix 1 according to RARECARE (van der Zwan JM, Trama A, Otter R, Larrañaga N, Tavilla A, Marcos-Gragera R, Dei Tos AP, Baudin E, Poston G, Links T; RARECARE WG. Rare neuroendocrine tumours: results of the surveillance of rare cancers in Europe project. Eur J Cancer. 2013 Jul;49(11):2565-78).
We selected all well, moderately and poorly differentiated, functioning and not functioning, gastroenteropancreatic neuroendocrine tumors, as well as thoracic and other sites neuroendocrine tumors (eg, lung, skin, thyroid, breast, etc).
Indeed, the register includes patients with small cell and large cell neuroendocrine carcinomas of the lung, who account for the vast majority of lung neuroendocrine neoplasms of our cohort.
Round 2
Reviewer 2 Report
The paper is improved. Several small points:
1. Simple summary: the first sentence is unclear. Please delete and begin with "The aim.."
2. Line 25 - of whom xx WERE taking insulin
3. Line 40 - "and"
4. Line 122-sources
5. Line 146 - protopathic bias - what is that?
6. Line 173 - "in" should be "on"
7. Line 193 - the math is wrong. Most were on insulin alone.... "The majority of those on insulin.".- do you mean on combination therapy???... "other glucose lowering medications " - what is that?
8. line 302 - "severity of disease" - do you mean duration?
Author Response
The paper is improved. Several small points:
- Simple summary: the first sentence is unclear. Please delete and begin with "The aim.."
Re: we agree and we delated the first sentece
- Line 25 - of whom xx WERE taking insulin
Re: we thank the reviewer for reporting. We corrected
- Line 40 - "and"
Re: thanks we corrected.
- Line 122-sources
Re: thanks for reporting mistake. We corrected
- Line 146 - protopathic bias - what is that?
Re: the protopathic bias occurs when symptoms treated by a drug are the manifestation of the yet-undiagnosed disease of interest (reverse causation)
- Line 173 - "in" should be "on"
Re: OK, changed.
- Line 193 - the math is wrong. Most were on insulin alone.... "The majority of those on insulin.".- do you mean on combination therapy???... "other glucose lowering medications " - what is that?
RE: we apologise for the mistake. The sentence referred to an older version of the table 1 and it should be a cancelled text. The numbers referred to those on combined treatment, most of them in treatment with metformin or sulfonylurea and the minority treated with other glucose lowering drugs. We removed the sentence that should be already cancelled.
- line 302 - "severity of disease" - do you mean duration?
Re: By the term "severity of disease" we mean poor glycaemic compensation, obviously both duration and severity contribute to increase the exposure in this context and, in many cases, uncompensated diabetes is also associated to a longer duration of the disease, as the reviewer pointed out. we have better reframed